# Prevention of SIVmac251 reservoir seeding in rhesus monkeys by early antiretroviral therapy

James B. Whitney[1,2], So-Yon Lim [1], Christa E. Osuna [1], Jessica L. Kublin[1], Elsa Chen[1], Gyeol Yoon[1], Po-Ting Liu [1], Peter Abbink [1], Erica N. Borducci[1], Alison Hill [3], Mark G. Lewis[4], Romas Geleziunas[5], Merlin L. Robb[6], Nelson L. Michael[7] & Dan H. Barouch[1,2]

The precise time when the viral reservoir is seeded during acute HIV-1 infection remains unclear. We previously demonstrated that the viral reservoir was seeded by day 3 following SIVmac251 infection in rhesus monkeys. Here we report the impact of initiating ART on day 0 (6 h), 1, 2, or 3 following intrarectal SIVmac251 infection in 20 rhesus monkeys (N = 5/ group). After 6 months of daily suppressive ART, antiretroviral drugs were discontinued, and viral rebound was monitored. 0% (0 of 5), 20% (1 of 5), 60% (3 of 5), and 100% (5 of 5) of animals that initiated ART on days 0 (6 h), 1, 2, or 3, respectively, showed viral rebound following ART discontinuation and correlated with integrated viral DNA in lymph node CD4+ T cells. These data demonstrate that the viral reservoir is seeded within the first few days of infection and that early ART initiation limits the viral reservoir.

[1] Center for Virology and Vaccine Research, Beth Israel Deaconess Medical Center, Harvard Medical School, Boston, MA 02215, USA. [2] Ragon Institute of MGH, MIT, and Harvard, Cambridge, MA 02139, USA. [3] Program for Evolutionary Dynamics, Harvard University, Cambridge, MA 02138, USA. [4] Bioqual, Rockville, MD 20852, USA. [5] Gilead Sciences, Foster City, CA 94404, USA. [6] U.S. Military HIV Research Program, Henry Jackson Foundation, 6720 Rockledge Drive, Bethesda, MD 20817, USA. [7] Walter Reed Army Institute of Research, 503 Robert Grant Ave, Silver Spring, MD 20910, USA. Correspondence and requests for materials should be addressed to J.B.W. (email: jwhitne2@bidmc.harvard.edu) or to D.H.B. (email: dbarouch@bidmc.harvard.edu)

The latent viral reservoir, which largely resides in memory CD4+ T cells of HIV-1-infected individuals, cannot be cleared solely by antiretroviral therapy (ART)[1–5]. A persistent archive of replication-competent virus remains in CD4+ T cells in HIV-1-infected individuals and is the source of rebound virus after the discontinuation of ART[6,7]. This persistent, latent HIV-1 reservoir represents a critical hurdle for HIV-1 eradication strategies[8,9].

We previously determined that the viral reservoir is seeded very early after mucosal SIVmac251 infection of rhesus monkeys by initiating ART on days 3, 7, 10, or 14[10]. We showed that the initiation of ART as early as day 3, prior to detectable viremia, was unable to prevent viral rebound following ART discontinuation despite 6 months of suppressive ART. Initiation of ART during acute HIV-1 infection in humans has similarly been shown to be unable to block establishment of the viral reservoir[11,12].

Very rapid initiation of ART following SIV challenge in monkeys as well as HIV-1 exposure in humans has been shown to block the establishment of chronic infection, but the dynamics of this process remain unclear. We therefore extended our previous study to assess the impact of initiating ART on days 0, 1, 2, or 3 following intrarectal SIVmac251 infection in rhesus monkeys to define more precisely the timing of seeding the latent viral reservoir in this model.

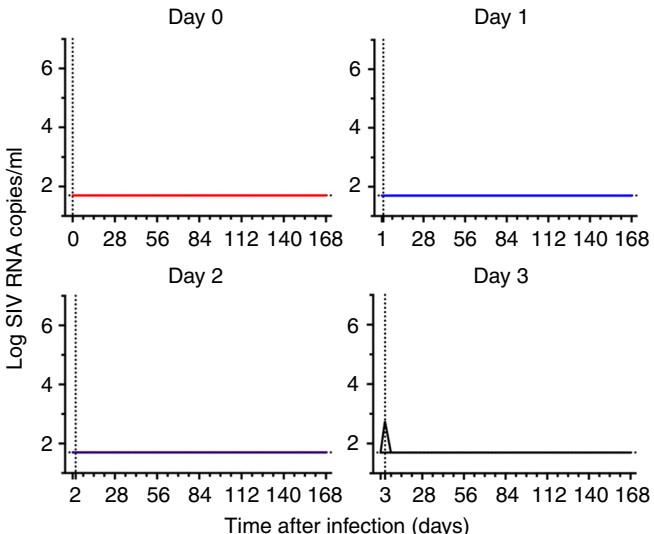

**Fig. 1** Plasma viral kinetics in SIV-infected rhesus monkeys following initiation of ART on days 0, 1, 2 and 3 post infection. Log plasma SIV RNA was monitored in rhesus monkeys following initiation of ART (N = 5/ group) on day 0 (6 h), day 1, day 2, and day 3 following intrarectal SIVmac251 infection. Assay limit of detection (<50 RNA copies/ml) and day of ART initiation are indicated with hatched lines

## Results

**Study design**. To assess the kinetics of seeding the SIVmac251 viral reservoir, we initiated suppressive antiretroviral therapy (ART) in 20 Indian-origin rhesus monkeys that were selected not to express the MHC class I alleles associated with spontaneous control of SIV replication (*Mamu-A\*01*, *-B\*08*, and *-B\*17*). 20 monkeys (N = 5/group) were infected with 500 tissue culture infective doses (TCID$_{50}$) of SIVmac251 and initiated ART on day 0 (6 h after infection), day 1, day 2, or day 3 following infection. The ART regimen consisted of daily s.q. injections of a preformulated cocktail of tenofovir, emtricitabine, and dolutegravir, as previously described[13]. Animals were treated with daily ART for 24 weeks, and plasma SIV RNA was monitored longitudinally every 2–4 weeks for the duration of the study. Prior to the initiation of ART, only a single animal had detectable blood viremia (572 SIV RNA copies/ml) on day 3 after infection. Following ART initiation, all animals had undetectable viral loads (<50 copies /ml) for the 24-week treatment period (Fig. 1).

**Detection of SIV DNA in tissue resident CD4+ T cells**. To assess the impact of early ART on the establishment of the viral reservoir, we evaluated cell-associated SIV DNA in peripheral blood mononuclear cells (PBMC), lymph node mononuclear cells (LNMC), and gastrointestinal mucosa mononuclear cells (GMMC) on the day of ART initiation and at week 24 (day 168) prior to ART discontinuation. We measured SIV DNA in CD4+ T cells from each anatomic compartment, as shown in Fig. 2. With the exception of one animal, SIV DNA was not detected in CD4+ T cells isolated from PBMC during this time period. Low levels of SIV DNA were detected in PBMC of the animal that showed SIV RNA on day 3 (Fig. 1) both prior to ART (2.46 log DNA copies/10$^6$ cells) and also at the time of ART cessation (1.57 log DNA copies/10$^6$ cells) (Fig. 2a).

In LNMCs, no SIV DNA was detected in animals that initiated ART on day 0 (6 h). However, we readily detected SIV DNA in LNMCs isolated from monkeys that started ART at later timepoints. At the time of ART initiation, DNA levels increased from day 1 (range, 0.88–1.99 log copies/10$^6$ cells) to day 2 (range, 1.62–2.2.36 log copies/10$^6$ cells) to day 3 (range, 2.09–3.21 log copies/10$^6$ cells) (Fig. 2b). In GMMCs, no SIV DNA was detected in animals that initiated ART on day 0. We detected SIV DNA in GMMCs in 3 of 5 animals (range, 0.85–2.29 log copies/10$^6$ cells) that initiated ART on day 1 and in all animals treated with ART on day 2 (range, 2.34–2.87 log copies/10$^6$ cells) or day 3 (range, 3.10–3.52 log copies/10$^6$ cells) (Fig. 2c). There was greater SIV DNA in LNMCs and GMMCs in animals that initiated ART on day 3 compared with animals that initiated ART on day 1 (LNMC, p = 0.0047; GMMC, p < 0.0001, Wilcoxon rank sum test). These data suggest that the viral reservoir was seeded within the first few days following infection.

**Early seeding of CD4+ T cell subsets**. Total CD4+ T cells were isolated from PBMCs. For GMMCs, CD4+ T subpopulations were sorted into naive (CD28+CD95−), and memory (CD95+). For LNMCs, CD4+ T cell subpopulations were sorted into naive, T follicular helper (T$_{FH}$, CD28+CD95+CXCR5+PD-1$^{high}$), T central and transitional memory (T$_{CM}$, CD28+CD95+, non-T$_{FH}$), and T effector and effector memory (T$_{EM}$) subsets as described in Methods. We compared SIV DNA in each cellular subpopulation among groups. Our results show differences in the frequency of SIV DNA in memory CD4+T subpopulations (T$_{CM}$ and T$_{EM}$) and T$_{FH}$ in LNMCs (Fig. 3a), as well as in CD4+T memory cells in GMMCs (Fig. 3b). Animals that initiated ART on days 0–2 demonstrated lower frequencies of SIV DNA in central and effector memory CD4+ T cell subpopulations in LNMCs and in memory CD4+T cells in GMMCs compared to animals that initiated ART on day 3.

We compared SIV DNA in memory cell subsets from different tissue compartments by normalizing SIV DNA copies per million cells of each subset to the proportion of each subset in terms of total cell equivalents. At the time of ART initiation, the T$_{CM}$ and naive populations harbored the majority of SIV DNA in LNMCs, whereas the T$_{FH}$ and T$_{EM}$ cells constituted a small proportion of the early SIV DNA burden (Supplementary Figure 1). SIV DNA decreased over time on suppressive ART in each cell subset.

We next quantified integrated SIV DNA in animals that initiated ART on day 1 or day 2 using an Alu-PCR assay as

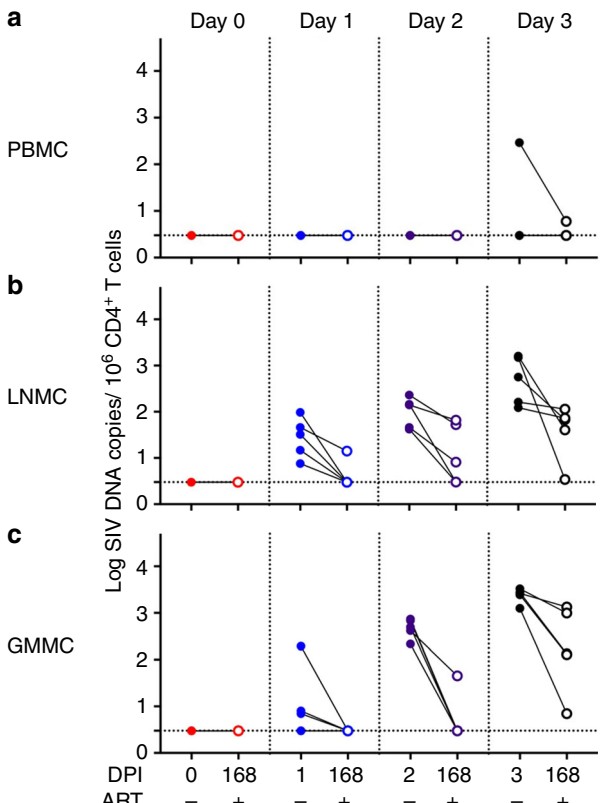

**Fig. 2** SIV DNA in blood and tissues pre-ART and at ART discontinuation. Cell-associated SIV DNA in **a** peripheral blood mononuclear cells (PBMC), **b** lymph node mononuclear cells (LNMC), and **c** gastrointestinal mucosa mononuclear cells (GMMC) in animals that initiated ART on days 0, 1, 2, and 3 following infection. Log SIV DNA copies/$10^6$ total CD4+ T cells is shown. Changes in levels of viral DNA prior to ART and at the time of ART discontinuation were compared using a Wilcoxon rank sum test. $p$ values are not significant unless they are indicated. Assay sensitivity, as indicated by the hatched line, is <3 copies/$10^6$ cells

described[14]. In PBMC, integrated SIV DNA was negative in these 10 monkeys (Supplementary Figure 2a). In GMMC, we detected integrated SIV DNA on the day of ART initiation in CD4+ T memory cells from one animal that initiated ART on day 1, while integrated DNA was detected in naive (two animals) and memory cells (three animals) from monkeys that initiated ART on day 2 (Supplementary Figure 2B). In LNMC, integrated SIV DNA was readily detected in all CD4+ T cell subpopulations. In the day 1 ART-treated animals, integrated SIV DNA was detected in $T_{CM}$ from four of five animals at the day of ART initiation, but it was observed in only one animal at the time of ART cessation; this animal subsequently showed viral rebound. In the day 2 ART treated animals, we detected integrated SIV DNA in naïve and $T_{EM}$ from 2 to 3 animals, in $T_{FH}$ from one animal, and in $T_{CM}$ from all five animals on the day of ART initiation. Integrated SIV DNA was also detected at the time of ART discontinuation, but only in the animals that showed viral rebound (Supplementary Figure 2c). The level of integrated SIV DNA was a median of 0.7–0.8 logs lower than total SIV DNA on both the day of ART initiation and the day of ART discontinuation.

We then compared the frequency of integrated SIV DNA in each CD4+ T cell subpopulation between animals that showed viral rebound ($n = 4$) vs. no viral rebound ($n = 6$) following ART discontinuation. Memory populations in GMMC from animals

that rebounded harbored significantly more integrated SIV DNA compared to animals that did not viral rebound. However, the frequency of integrated SIV DNA detected in memory populations decreased substantially by the day of ART discontinuation (Fig. 4a). In LNMC, total SIV DNA was detected in all CD4+ T cell subpopulations from 33 to 100% of animals. On the day of ART discontinuation, we found that $T_{CM}$ was the sole subpopulation that harbored detectable integrated SIV DNA in all animals that rebounded ($n = 4$). Moreover, the frequency of integrated SIV DNA detected in $T_{CM}$ was significantly different between animals that rebounded compared with those that did not rebound following ART discontinuation ($p = 0.048$, Fisher's exact test) (Fig. 4b).

**Viral rebound after ART discontinuation**. At week 24, ART was discontinued in all animals, and plasma SIV RNA was monitored for evidence of viral rebound, which was defined as plasma SIV RNA > 50 copies/ml. Viral rebound was observed during the follow-up period in 100% (5 of 5), 60% (3 of 5), 20% (1 of 5), and 0% (0 of 5) of animals that initiated ART on days 3, 2, 1, and 0 (6 h), respectively (Fig. 5a). Viral rebound occurred on days 10–21 following ART discontinuation. Animals did not seroconvert until viral rebound.

The Kaplan–Meier curve of viral rebound in each group also demonstrate a significant reduction in rebound with the initiation of earlier ART (Fig. 5b, $p = 0.0016$, Log-rank test). However, we did not detect a significant delay in the time to viral rebound for any group.

To assess for potential genetic signatures of rebound virus, we performed single genome sequencing using viral RNA isolated from peak rebound viremia (days 28–42 following ART discontinuation), as described in the Methods. These data show that *env* sequences of rebound virus were very similar to the challenge stock (Supplementary Figure 3a). We identified six nonsynonymous changes present only in the rebound virus (Supplementary Figure 3b), of which five were in gp41. In one animal that initiated ART on day 1, two mutations (99R, 709M) were identified. In two animals that initiated ART on day 2, nonsynonomous changes (837A, 874T) were observed. In animals that initiated ART on day 3, 99R and 863E mutations were found. The minimal mutations found in rebound virus suggest that little to no virus replication occurred prior to seeding the viral reservoir.

**Viral dynamics modeling**. We combined data from this study with the previous study[10] in which ART was initiated on days 3, 7, 10 or 14 to evaluate the probability, timing, and kinetics of viral rebound, as described in the Supplemental Methods. We calculated AUC viremia before ART for animals that showed detectable viremia and estimated it for those that did not using a simple model of viral growth calibrated to the population. Consistent with the model of viral load-dependent reservoir seeding[11], the pre-ART AUC viral load was highly correlated with the eventual reservoir size at the time of ART interruption as measured by SIV DNA, particularly in PBMC (Supplementary Figure 4a). The probability of viral rebound as a function of reservoir size was also examined using a simple function predicted by previous analysis[15]. Total or integrated SIV DNA in PBMC was one predictor of viral rebound, but pre-ART AUC viral load was a more robust predictor of viral rebound (Supplementary Figure 4b).

**Discussion**
In this study, we demonstrate the impact of very early treatment with ART following SIVmac251 infection in rhesus monkeys.

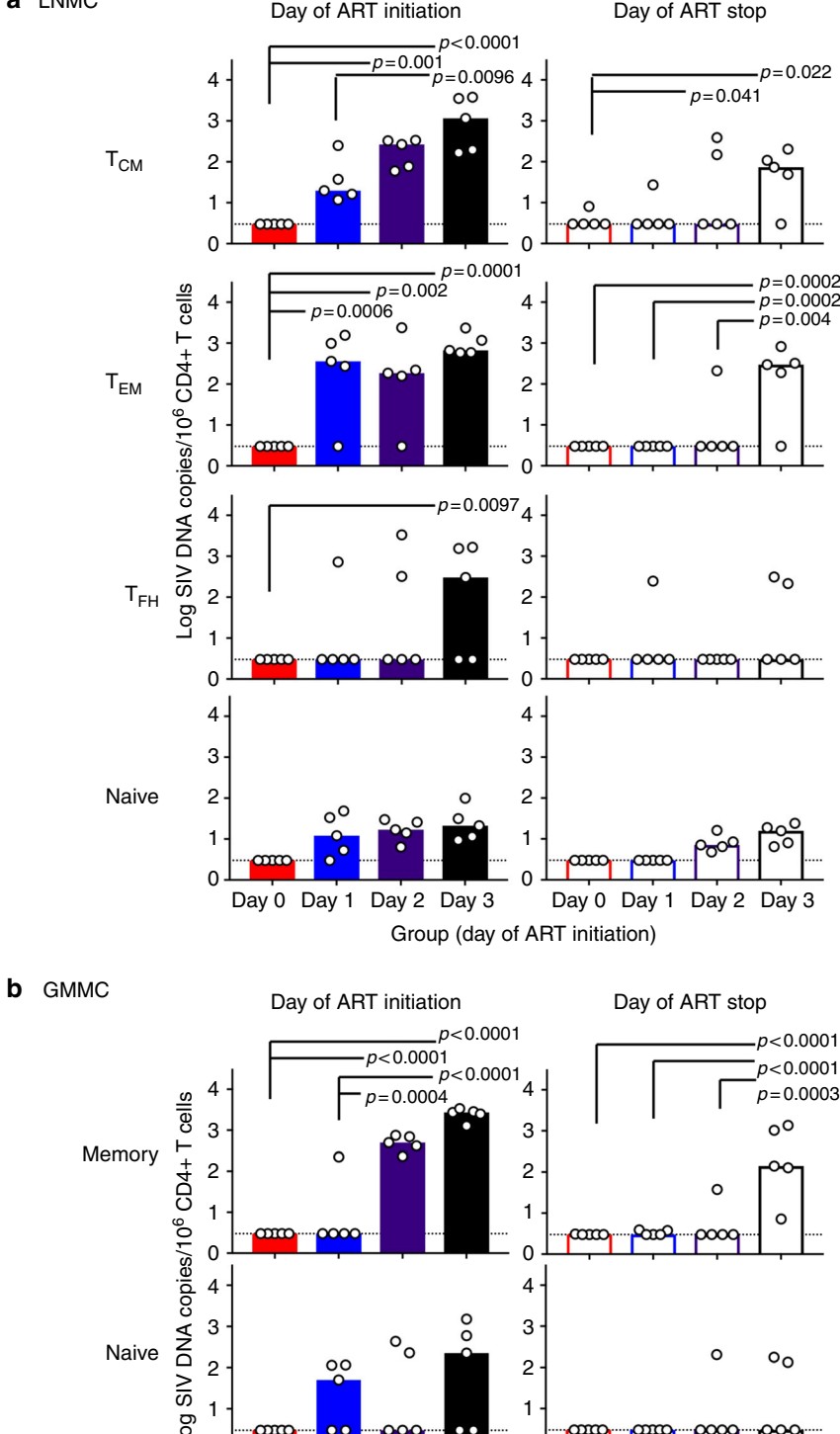

**Fig. 3** Comparison of SIV DNA in CD4+ memory T cell subsets. Mononuclear cells were isolated from tissues of animals that initiated ART on days 0, 1, 2 and 3 following infection and sorted into populations of CD4+ T cells: **a** lymph node naïve, central memory ($T_{CM}$), follicular helper T cells ($T_{FH}$) and effector memory ($T_{EM}$); **b** colon naive and memory. SIV DNA is expressed as log copies/$10^6$ CD4+ T cells at the time of ART initiation and stopping ART. All data points are shown with the median expressed as bars. $p$ values reflect $t$-tests that were used to compare means of proviral DNA in each subset between each group

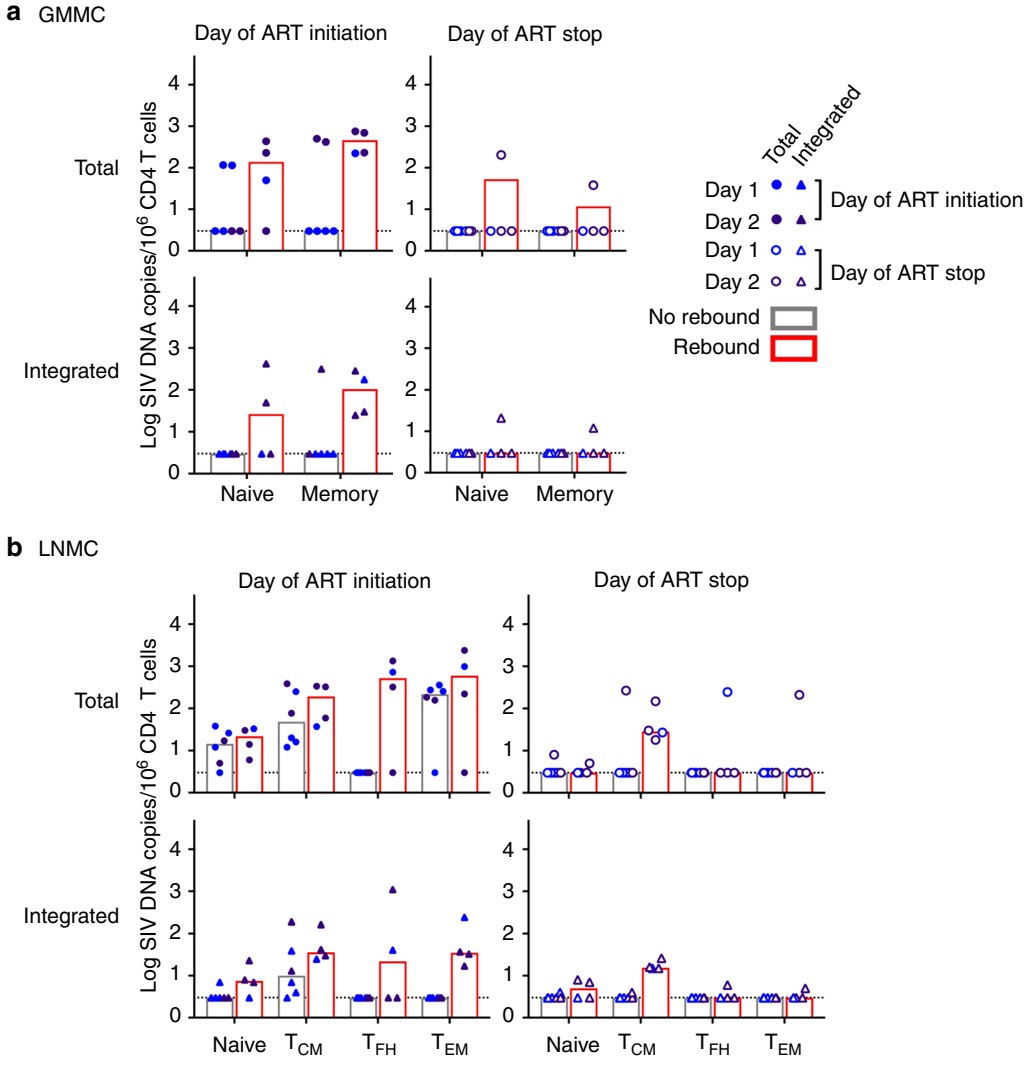

**Fig. 4** Comparison of integrated SIV DNA in CD4+ T cell subsets following ART discontinuation. **a** Total and integrated levels of SIV DNA in sorted naive and memory CD4+ T cells in GMMC and **b** naive, central memory ($T_{CM}$), follicular helper ($T_{FH}$) and effector and effector memory ($T_{EM}$) CD4+ T cells in LNMC are shown. Ten animals that initiated ART on day 1 ($n = 5$) or 2 ($n = 5$) were divided into two groups: animals that did not rebound ($n = 6$) and animals that did rebound ($n = 4$) following ART withdrawal. All data points are shown. Bars indicate the median of values. $p$ values reflect Fisher's exact test to compare the frequency of the SIV DNA positive sample distribution between each group

ART reliably prevented the establishment of persistent infection when initiated at 6 h following exposure, but ART rapidly lost efficacy when initiated on days 1–2 and provided no efficacy when initiated on day 3. These data demonstrate that the viral reservoir is seeded remarkably early in this model, generally within 1–2 days following exposure. These data have important implications for understanding and improving HIV-1 therapeutic and cure strategies.

These data confirm and extend our prior findings that the viral reservoir was seeded within 3 days following SIVmac251 infection of rhesus monkeys[10]. In this prior study, we assessed the impact of ART initiation on day 3, 7, 10, or 14. In the present study, we evaluated the efficacy of ART initiation on day 0 (6 h), 1, 2, or 3 using the same experimental model, challenge virus, and ART duration. Importantly, similar results were observed in the day 3 treatment groups from both studies. Collectively, these two studies demonstrate that early ART reduced the size of the viral reservoir in rhesus monkeys, consistent with findings in HIV-1-infected humans[16], but that viral rebound still occurred routinely following ART discontinuation unless ART was initiated within

the first 2 days of infection. We cannot exclude the possibility that an extended period of ART suppression >6 months may reduce the frequency of viral rebound in the early treatment groups.

Our data also extend prior studies demonstrating the efficacy of blocking infection in monkeys when ART was initiated rapidly following SIV challenge[17,18]. Our data show that initiation of ART within 6 h of exposure effectively blocks establishment of SIVmac251 infection. Partial efficacy is evident at 1–2 days, but no efficacy was observed at 3 days. Potential implications of these data to post-exposure prophylaxis in humans, however, remain to be determined.

We observed SIV DNA in lymph nodes in all animals that initiated ART on days 1–3 (Fig. 2). The fact that a subset of these animals did not rebound following ART discontinuation suggests the existence of a labile population of CD4+ T cells containing SIV DNA that may not develop into a permanent viral reservoir. Indeed, only animals that showed persistence of integrated SIV DNA in CD4+T cells (mostly $T_{CM}$ cells) in lymph nodes at the time of ART discontinuation demonstrated viral rebound (Fig. 4b). These data are consistent with studies from our

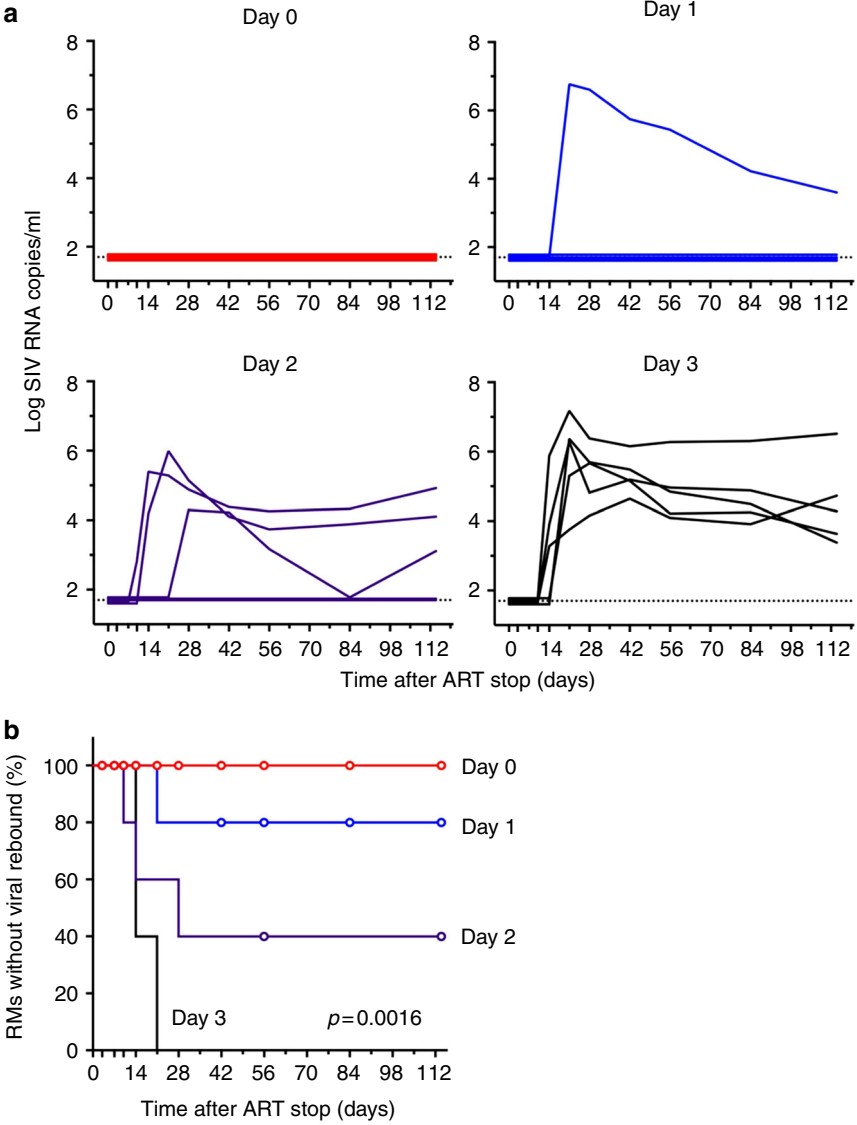

**Fig. 5** Viral rebound kinetics following ART discontinuation. **a** ART was discontinued after 24 weeks, and plasma SIV RNA was assessed on days 0, 3, 7, 10, 14, and then weekly. **b** Kaplan–Meier curve showing the time to viral rebound for each group of animals. The differences between groups were analyzed by the log-rank test and *p* value is indicated

laboratory and others in which early foci of presumably labile virus were detected in the first few days following SHIV infection despite the administration of a fully protective dose of a broadly neutralizing antibody[19,20].

The dynamics of reservoir seeding pre-ART and viral rebound post-ART have been the focus of extensive viral dynamics models but relatively few experimental studies. By analyzing monkeys initiating ART between 6 h and 14 days of infection[10], we show that the pre-ART AUC viral loads (inferred or observed) were highly correlated with the size of the SIV DNA reservoir following ART suppression, although not directly proportional (slope of ~0.6 vs 1 on a log-log scale)[11,21]. These findings agree with data from humans in which the reservoir was measured by quantitative viral outgrowth assays[11]. In addition, we observed that a simple model[15] described the relationship between reservoir size at ART cessation and the probability of rebound, but that the relationship was more robust for pre-ART AUC viral loads than for SIV DNA (Supplementary Figure 5). The probability of no rebound increased more rapidly than expected as DNA levels declined[10]. This model also predicts that reservoir

sizes small enough to avoid rebound in a subset of animals may lead to rebound after a long delay in other animals, as has been observed in humans[22,23].

Our data show that SIVmac251 seeds the viral reservoir generally within 1–3 days following intrarectal infection in rhesus monkeys and resists 6 months of suppressive ART. A subset of early SIV DNA-containing CD4+ T cells, however, may be labile and may not result in a persistent reservoir. An increased understanding of the virologic parameters that predict early reservoir seeding and viral rebound following ART discontinuation will help guide future HIV-1 eradication efforts.

## Methods

**Study design**. These experiments were designed to investigate whether early administration of ART would restrict the establishment of the reservoir in SIV-infected rhesus monkeys. Twenty Indian-origin, outbred, young adult, male specific pathogen-free (SPF) rhesus monkeys (*Macaca mulatta*) that did not express the class I alleles *Mamu-A\*01*, *Mamu-B\*08*, and *Mamu-B\*17* associated with enhanced virologic control were housed at Bioqual, Inc. Animals used in this study were infected with 500 tissue culture infective doses (TCID_{50}) of SIVmac251, essentially as described (*10*). Animals were balanced by body weight and assigned to each

group ($N = 5$/group). Monkeys were maintained according to the guidelines of the *NIH Guide for the Care and Use of Laboratory Animals*. The animal studies described were approved by the Bioqual Institutional Animal Care and Use Committee (IACUC).

**ART regimen**. The pre-formulated antiretroviral therapy (ART) cocktail containing two reverse transcriptase inhibitors, 5 mg/mL tenofovir disoproxil fumarate (TDF) and 40 mg/mL emtricitabine (FTC), plus 2.5 mg/mL of the integrase inhibitor dolutegravir (DTG) was administered once daily at 1 mL/kg body weight via the subcutaneous route[13]. This ART regimen was initiated on days 0 (6 h), 1, 2, and 3 post infection and continued for 24 weeks. Monkeys were bled up to two times per week for study week 1 and 2, monthly by study week 24, and then weekly until study week 40 to monitor viral rebound. Tissues including lymph node and colorectal mucosa were biopsied prior to initiation of ART and at week 24 to measure cell-associated SIV DNA.

**SIV DNA assay**. Total cellular DNA was isolated from total or sorted CD4+ T cells using a QIAamp DNA Blood Mini kit (Qiagen) and tissue-specific proviral DNA was quantitated as previously reported. Isolated total DNA quality was verified by average $A_{260}/A_{280}$ ratio of 1.85 (range 1.77–1.96). The absolute quantification of viral DNA in each sample was carried out on an ABI 7300 Real-Time PCR system (Applied Biosystems) using primers specific to a conserved region SIVmac239. Primer sequences that used in the study were forward primer s-Gag-F: 5′-GTCTGCGTCATCTGGTGCATTC-3′, reverse primer s-Gag-R: 5′-CACTAGG TGTCTCTGCACTATCTGTTTTG-3′, and the probe s-Gag-P: 5′-CTTCCTCAG TGTGTTTCACTTTCTCTTCTGCG-3′, linked to Fam and BHQ (Biosearch Technologies, Petaluma, CA). For each qPCR, triplicate wells were set up for both SIV and GAPDH and all samples were directly compared to a linear SIV standard and the simultaneous amplification of a fragment of human GAPDH gene. The sensitivity of linear standards was compared against the 3D8 cell line as a reference standard as described[3]. All PCR assays were performed using 100–200 ng of sample DNA.

**Integrated SIV DNA assay**. Genomic DNA was prepared from sorted CD4+ T cell subsets using either a QIAamp DNA blood mini kit or QIAamp DNA micro kit following Manufacturer's instruction. To detect integrated SIV DNA, we applied the method described by Nishimura et al. with modification for use of rhesus macaque specific Alu sequences[14]. A cloned cell line, 3D8 containing a single integrated copy of SIV DNA was used an integration standard. Genomic DNA was serially diluted in 10-fold dilutions (1000 to 1 cell equivalents) and applied to the first-round PCR. SIV LTR nested PCR products were then subjected to a standard qRT-PCR. Cell numbers analyzed in each reaction was confirmed by simultaneous qPCR of GAPDH genes.

**Flow cytometric sorting of CD4+ T cell memory subsets**. Single cell suspensions were prepared from lymph node and gastrointestinal mucosal biopsies as previously described [10] and stored in liquid nitrogen. Cryopreserved samples of PBMC, LNMC and GMMC were thawed at 37 °C in RPMI 1640 containing 10% FBS and benzonase (Millipore) at 50 U/mL. Cells were re-suspended in 1X PBS containing Aqua LIVE/DEAD Fixable Dead Cell Stain (Life Technologies) for 20 min at RT in the dark. Cells were washed and re-suspended in wash buffer containing the following fluorescently conjugated antibodies at the indicated dilutions: CD45 (D058-1283, BD #563861,1:80), CD28 (CD28.2, BD #5622596, 1:200), CD4 (L200, BD #563913, 1:100), CCR7 (150503, BD # 561271, 1:40), CD95 (DX2, BD #555674, 1:100), CD3 (SP34-2, BD # 557917, 1:40), and CD8 (SK1, BD #560179, 1:80), CXCR5 (MU5UBEE, Thermo-Fisher # 25-9185-42, 1:30), and PD-1 (EH12.2H7, Biolegend # 329919, 1:100). Cells were sorted in purity mode, using a BD FACSAria II. Sorted CD4+ T cells were FSC singlets, live, CD45+CD3+ CD4+CD8− lymphocytes and subsets were defined as follows. For GMMC: naïve (CD28+CD95−) and memory (CD95+). For LNMC: naive (CD95−CD28+), T follicular helper cells (Tfh, CD28+CD95+CXCR5+PD-1$^{high}$), central memory (CD95+CD28+, non-Tfh), effector and effector memory (CD95+CD28−). CXCR5+ events were determined by a CXCR5 FMO antibody stained sample.

**Statistical analyses**. Comparisons of changes in values between prior to and after ART within groups were analyzed by use of a Wilcoxon rank sum test. Analyses of SIV DNA comparison in each CD4+ T cell subset between groups at either study entry (ART initiation) or ART stop were performed using either a Kruskal–Wallis test or unpaired, equal variance, one-tailed t-tests. Transformed log10 SIV RNA levels in plasma were calculated using GraphPad Prism (version 6.0). $p$ values were adjusted for multiple comparisons when more than two groups were compared.

**Disclaimer**. The views expressed in this manuscript are those of the authors and do not represent the official views of the Department of the Army or the Department of Defense.

## Data availability

All data are freely available in this manuscript and upon request.

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

## Acknowledgements

We thank J. Hesselgesser, J. Harrison, C. Gittens, J. Yalley-Ogunro, H. Anderson, and W. Wagner for expert animal husbandry and care. We acknowledge support from the US Army Medical Research and Materiel Command and the US Military HIV

Research Program, Walter Reed Army Institute of Research through its cooperative agreement with the Henry M. Jackson Foundation for the Advancement of Military Medicine (W81XWH-11-2-0174); the NIH (AI091514, AI096040, AI122942, AI124377, AI126603, AI127089, AI128751, AI131365); and the Ragon Institute of MGH, MIT, and Harvard.

## Author contributions

D.H.B., N.L.M., M.L.R., and J.B.W. designed the studies. C.E.O., J.L.K., E.C., G.Y., S.Y.L., and P.A. led the virologic assays. C.E.O., J.L.K., E.C., G.Y., and S.Y.L. led the flow cytometric sorting. A.L.H. led the mathematical modeling. P.L. and S.Y.L. led the viral sequencing. E.N.B., J.B.W., and M.G.L. led the animal studies and specimen processing. R.G. provided the antiretroviral drugs. J.B.W. and D.H.B. led the studies and wrote the paper with all co-authors.

## Additional information

**Competing interests:** The authors declare no competing interests.

