## [Peer Review File · Nature Communications]

Reviewers' Comments:

Reviewer #1:

Remarks to the Author:

This is a well-written manuscript on the effects of early antiviral therapy initiated within the first 3 days after SIVmac251 inoculation. It is an extension of earlier work by these authors. The manuscript can be improved by the following suggestions. The overall conclusions are that seeding of viral reservoirs occurs very rapidly after exposure, and that PEP is most likely to be effective when initiated within the first day after exposure.

I have only a few suggestions for improvement:

- Page 6, the statement "...showed that the median log AUC was reduced in viremic animals that initiated ART on days 0-1 compared to..." should be rephrased. As only 1 of the "days 0-1" animals became infected, calculating a median of the viremic animals (in this case just 1 animal) doesn't make sense. Instead, figure 4B indicates that the 1 animal of day 1 that had break-through infection had similar AUC as the break-through animals of groups "day 2 and 3".

- Do the authors have any data on when animals seroconverted, as such information would be of interest also for human studies. In particular, it would be good to know whether animals that became infected already made some antibodies prior to ART withdrawal, or if they seroconverted only after ART withdrawal. And the animals that were not viremic, did they have any antibody response (as prior PEP studies in macaques, including infant macaques, have sometimes found transient detection of low-levels of antibodies during the first weeks after exposure, which then became undetectable). While this information won't change the conclusions of this study, it would be of scientific interest.

Reviewer #2:

Remarks to the Author:

This manuscript by Whitney et al. explores whether early initiation of antiretroviral therapy has an influence in the establishment of the SIV latent reservoir as a model for post-exposure prophylaxis (PEP). For that, male rhesus monkeys are infected intra-rectally with a single dose of the lab adapted strain of SIV, SIVmac251. After exposure, ART is administered 6 hours or 1, 2 or 3 days post-challenge during 24 weeks. At that time, ART is discontinued to assess viral rebound. The levels of SIV DNA are analyzed in different CD4 subsets prior ART and after 24 weeks of ART in lymph node mononuclear cells (LMNC) and gastrointestinal mucosa mononuclear cells (GMNC). This work complements previous work from the same group published in Nature in 2014. In this previous work, the authors follow a similar protocol of infection but rhesus are treated at later time points after inoculation. It is worth notice that both studies, the one published in Nature and the one presented in this manuscript, are limited to intra-rectally infection of male with a single dose of SIV. This may influence the evaluation of how effective PEP will be to inhibit the establishment of the latent reservoir in the clinical setting. It will be suggested to emphasize this limitation in the discussion section. In spite of this, the results presented in this manuscript are important in the field of HIV therapy, both for PEP as well as for cure strategies.

There are a few suggestions that the authors should also try to address before publication:

- In spite of the lack of detection of SIV RNA in blood, the authors find SIV DNA in CD4 T cells in both the LMNC and the GMNC one day and beyond after infection. These levels are drastically reduced after

the 24-week treatment with ART. The authors conclude that to be due to "the existence of a labile population of CD4 T cells containing SIV DNA that may not develop into a permanent reservoir". However, the data could also be interpreted as cells that are productively infected and die after initial infection and not be part of the latent reservoir. It will be important to address whether SIV RNA can be detected that early in those cell compartments and whether the treatment reduces SIV RNA levels as well as SIV DNA. Lack of SIV RNA detection will definitely prove the presence of a labile reservoir.

- The analysis of SIV DNA in different subsets is confusing. First, there is not data about PBMCs even though it is mention in the text. However, there is not SIV detected in these samples, so it probably needs to be just removed from the text. Different subsets are isolated from LNMC and GNMC but it is not clearly stated in the text. Were the different subsets analyzed in LNMCs analyzed also in GNMCs? What is the memory phenotype of the cells isolated in GNMCs?

Minor comments:

- It is not completely clear what is the meaning of the analysis of env sequences. The lack of accumulation of mutations suggest that little to no replication is needed to establish the latent reservoir in this model as the mutations detected could be accumulated post-ART interruption. This result may suggest that the inoculum is able to establish the latent reservoir directly. No discussion whatsoever is including about what are the "implications for understanding and improving PEP and cure strategies" based on these results.

Reviewer #3:

Remarks to the Author:

This is an interesting manuscript extending the finding of this group and others that virus rebounds in SIV-infected macaques even after very early initiation of ART (Whitney Nature 2014). Here the authors extend their previous work by administering ART within 6 hours of infection. Unfortunately they only maintain ART for 6 months and this could be highly relevant to determine if there is an unstable reservoir established early that will decay on ART. Six months may not be long enough. They find that there is no DNA detected in tissue or blood in animals who start ART on day 0 and none of these animals rebound post cessation of ART. The frequency of rebound was greater with delays in ART initiation. The AUC post viral rebound correlated with DNA at the time of ART cessation. Using a mathematical model that included data from their previous publication, they found that AUC of VL pre ART was a better predictor of rebound. The model leads to a very similar conclusion as their previous paper. They conclude that there is a narrow window of time of 6 hours for PEP to prevent establishment of the reservoir.

The study is well done and systematically analysed and extends previous findings on when the viral reservoir is established. I don't believe their findings can be translated to recommendations regarding PREP. There have been multiple studies published that address this question, in relation to PEP (see Irvine CID 2015 for a meta analysis). The authors argue that this work is informative for post exposure prophylaxis (PEP) which I disagree with and the discussion in relation to this is potentially misleading.

The novelty in this study is defining in detail the kinetics and location of establishing a long term reservoir, not whether PEP recommendations work or not. Unfortunately, they provide no mechanistic insights into why some monkeys who have DNA detected in tissue, don't rebound nor do they extend our understanding of whether there is a labile and fixed reservoir.

Is the work convincing, and if not, what further evidence would be required to strengthen the conclusions?

Specific comments

Introduction

1. I disagree with the authors that this work implies there is a 'narrow window of time during which PEP is reliably effective prior to seeding the latent viral reservoir'. This work would suggest that the window is 6 hours, which is clearly not the case clinically in human infection. Recommendations for PEP have been routinely with a window for initiation of 72 hours and administration for 28 days, with extremely rare reports of seroconversion. A recent meta-analysis of PEP in NHPs [Irvine CID 2015] is probably more informative in relation to optimal timing of PEP than this small non-randomised study. Any discussion related to efficacy of PEP, should be removed as the study was not designed to answer this question.

Results

1. Figure 1 is not terribly informative and could be moved to supplemental data
2. Seeding of the reservoir and the virological events in tissue that are completed prior to initiation of very early ART are a very interesting aspect of this study, that could never be performed in humans. The authors refer to this as there potentially being a labile and stable reservoir. However, the investigators only measure HIV DNA, a crude marker of the reservoir which includes unintegrated, integrated and 2LTR circular DNA. Unfortunately no investigations were made to better define this labile reservoir. Further work should be done on the SIV DNA+ samples which should include integrated DNA and cell associated RNA as a minimum to better understand this very important issue. An assessment of the proportion of intact virus would also be highly informative
3. Unfortunately, a very short duration of ART was given of only 6 months. Do the authors think that a longer duration may have inhibited viral rebound?
4. The relationship between AUC following viral rebound and time of ART initiation is interesting, especially because the time to rebound did not vary in the 4 groups. In addition, AUC after viral rebound correlated with SIV DNA in tissue memory CD4+ T-cells at the time of ART discontinuation. Looking at the raw data in figure 4A, it seems that peak viral load was similar for all groups (except day 0 where there was no viral rebound) and set point also pretty similar, except for one monkey in the day 2 group. I am concerned that this relationship is falsely skewed by monkeys that don't rebound (for whom there was a wide range of SIV DNA in tissue as shown in figure S2). Was there still a relationship between AUC post viral rebound and tissue DNA at the time of ART interruption, if the analysis only included monkeys who rebounded? I wonder if this might be a more meaningful analysis
5. An extremely important aspect of the study is understanding why monkeys with detectable DNA don't rebound and further mechanistic work should be performed to understand this.
6. It seems that naïve T cells were infected by 24 hours in both LN and GMMC (figure 3B) and in central memory cells in LN (figure S1). This would imply direct infection of these cells, rather than infection of activated memory cells that revert to a naïve or central memory state. Furthermore, it seems that these infected cells are less stable in GMMC compared to LN (figure 3 and S1) This should be further discussed in relation to establishment of long lived infection in different T-cell subsets and in different tissue sites.

Discussion

1. On page 9, the authors talk about 'initial foci of virus in distal tissues'. I was unclear what this meant. If there were a subset of monkeys in whom DNA was detected in tissue but there was no viral rebound, this study provided the opportunity to better characterise this labile reservoir
2. The concluding statement in relation to PEP is misleading and should be removed from the manuscript

Reviewer #1 (Remarks to the Author):

This is a well-written manuscript on the effects of early antiviral therapy initiated within the first 3 days after SIVmac251 inoculation. It is an extension of earlier work by these authors. The manuscript can be improved by the following suggestions. The overall conclusions are that seeding of viral reservoirs occurs very rapidly after exposure, and that PEP is most likely to be effective when initiated within the first day after exposure.

We thank the reviewer for the positive comments.

I have only a few suggestions for improvement:

- Page 6, the statement "...showed that the median log AUC was reduced in viremic animals that initiated ART on days 0-1 compared to...." should be rephrased. As only 1 of the "days 0-1" animals became infected, calculating a median of the viremic animals (in this case just 1 animal) doesn't make sense. Instead, figure 4B indicates that the 1 animal of day 1 that had break-through infection had similar AUC as the break-through animals of groups "day 2 and 3".

We thank the reviewer for noting this error. We have modified the statement on page 7 of the revised manuscript accordingly.

- Do the authors have any data on when animals seroconverted, as such information would be of interest also for human studies. In particular, it would be good to know whether animals that became infected already made some antibodies prior to ART withdrawal, or if they seroconverted only after ART withdrawal. And the animals that were not viremic, did they have any antibody response (as prior PEP studies in macaques, including infant macaques, have sometimes found transient detection of low-levels of antibodies during the first weeks after exposure, which then became undetectable). While this information won't change the conclusions of this study, it would be of scientific interest.

The animals did not seroconvert prior to ART withdrawal and viral rebound. This is added on page 7 of the revised manuscript.

Reviewer #2 (Remarks to the Author):

This manuscript by Whitney et al. explores whether early initiation of antiretroviral therapy has an influence in the establishment of the SIV latent reservoir as a model for post-exposure prophylaxis (PEP). For that, male rhesus monkeys are infected intra-rectally with a single dose of the lab adapted strain of SIV, SIVmac251. After exposure, ART is administered 6 hours or 1, 2 or 3 days post-challenge during 24 weeks. At that time, ART is discontinued to assess viral rebound. The levels of SIV DNA are analyzed in different CD4 subsets prior ART and after 24 weeks of ART in lymph node mononuclear cells (LMNC) and gastrointestinal mucosa mononuclear cells (GMNC). This work complements previous work from the same group published in Nature in 2014. In this previous work, the authors follow a similar protocol of infection but rhesus are treated at later time points after inoculation. It is worth notice that both studies, the one published in Nature and the one presented in this

manuscript, are limited to intra-rectally infection of male with a single dose of SIV. This may influence the evaluation of how effective PEP will be to inhibit the establishment of the latent reservoir in the clinical setting. It will be suggested to emphasize this limitation in the discussion section. In spite of this, the results presented in this manuscript are important in the field of HIV therapy, both for PEP as well as for cure strategies.

We thank the reviewer for the positive comments. We have included caveats related to the implications for human PEP strategies on pages 9-10 of the discussion.

There are a few suggestions that the authors should also try to address before publication:

- In spite of the lack of detection of SIV RNA in blood, the authors find SIV DNA in CD4 T cells in both the LNMC and the GNMC one day and beyond after infection. These levels are drastically reduced after the 24-week treatment with ART. The authors conclude that to be due to “the existence of a labile population of CD4 T cells containing SIV DNA that may not develop into a permanent reservoir”. However, the data could also be interpreted as cells that are productively infected and die after initial infection and not be part of the latent reservoir. It will be important to address whether SIV RNA can be detected that early in those cell compartments and whether the treatment reduces SIV RNA levels as well as SIV DNA. Lack of SIV RNA detection will definitely prove the presence of a labile reservoir.

We appreciate the reviewers’ comments. We did not have sufficient cells to assess cellular SIV RNA. Instead, we assessed integrated SIV DNA in PBMC, LNMC, and GMMC both early and after the 24 week treatment with ART. Integrated DNA after long-term ART suggested the persistent reservoir. Integrated DNA after long-term ART in CD4+ T cells (mostly T_{CM} cells) in lymph nodes correlated with viral rebound following ART discontinuation. This has been added in Figure 4 and Supplementary Figure S2, and detailed on pages 6-7 of the revised manuscript.

- The analysis of SIV DNA in different subsets is confusing. First, there is not data about PBMCs even though it is mention in the text. However, there is not SIV detected in these samples, so it probably needs to be just removed from the text. Different subsets are isolated from LNMC and GNMC but it is not clearly stated in the text. Were the different subsets analyzed in LNMCs analyzed also in GNMCs? What is the memory phenotype of the cells isolated in GNMCs?

We have clarified in the revised manuscript that we were unable to sort all the subpopulations from GMMC due to limited cell numbers obtained from the pinch biopsies. We therefore only have naïve and total memory CD4+ T cells from GMMC in Fig. 4A and Supplementary Figures S1 and S2B of the revised manuscript. In contrast, we were able to sort and to analyze more CD4+ T cell subpopulations from LNMC.

Minor comments:

- It is not completely clear what is the meaning of the analysis of env sequences. The lack of accumulation of mutations suggest that little to no replication is needed to establish the latent reservoir in this model as the mutations detected could be accumulated post-ART interruption.

This result may suggest that the inoculum is able to establish the latent reservoir directly. No discussion whatsoever is including about what are the “implications for understanding and improving PEP and cure strategies” based on these results.

We have added this point on page 8 of the revised manuscript.

Reviewer #3 (Remarks to the Author):

This is an interesting manuscript extending the finding of this group and others that virus rebounds in SIV-infected macaques even after very early initiation of ART (Whitney Nature 2014). Here the authors extend their previous work by administering ART within 6 hours of infection. Unfortunately they only maintain ART for 6 months and this could be highly relevant to determine if there is an unstable reservoir established early that will decay on ART. Six months may not be long enough. They find that there is no DNA detected in tissue or blood in animals who start ART on day 0 and none of these animals rebound post cessation of ART. The frequency of rebound was greater with delays in ART initiation. The AUC post viral rebound correlated with DNA at the time of ART cessation. Using a mathematical model that included data from their previous publication, they found that AUC of VL pre ART was a better predictor of rebound. The model leads to a very similar conclusion as their previous paper. They conclude that there is a narrow window of time of 6 hours for PEP to prevent establishment of the reservoir.

The study is well done and systematically analysed and extends previous findings on when the viral reservoir is established. I don't believe their findings can be translated to recommendations regarding PREP. There have been multiple studies published that address this question, in relation to PEP (see Irvine CID 2015 for a meta analysis). The authors argue that this work is informative for post exposure prophylaxis (PEP) which I disagree with and the discussion in relation to this is potentially misleading.

The novelty in this study is defining in detail the kinetics and location of establishing a long term reservoir, not whether PEP recommendations work or not. Unfortunately, they provide no mechanistic insights into why some monkeys who have DNA detected in tissue, don't rebound nor do they extend our understanding of whether there is a labile and fixed reservoir.

We thank the reviewer for the overall positive comments. We have removed the discussion regarding PEP in humans. As detailed below, we have included integrated SIV DNA data. Integrated SIV DNA in LNMC at the time of ART discontinuation correlated with viral rebound.

Is the work convincing, and if not, what further evidence would be required to strengthen the conclusions?

Specific comments

Introduction

1. I disagree with the authors that this work implies there is a ‘narrow window of time during which PEP is reliably effective prior to seeding the latent viral reservoir’. This work would

suggest that the window is 6 hours, which is clearly not the case clinically in human infection. Recommendations for PEP have been routinely with a window for initiation of 72 hours and administration for 28 days, with extremely rare reports of seroconversion. A recent meta-analysis of PEP in NHPs [Irvine CID 2015] is probably more informative in relation to optimal timing of PEP than this small non-randomised study. Any discussion related to efficacy of PEP, should be removed as the study was not designed to answer this question.

As suggested, we have removed the comparisons with PEP in humans.

Results

1. Figure 1 is not terribly informative and could be moved to supplemental data.

We prefer to keep Figure 1 in the manuscript, because the negative viral loads prior to and during ART suppression are critical. If space is limited, then we can move this figure to supplemental data.

2. Seeding of the reservoir and the virological events in tissue that are completed prior to initiation of very early ART are a very interesting aspect of this study, that could never be performed in humans. The authors refer to this as there potentially being a labile and stable reservoir. However, the investigators only measure HIV DNA, a crude marker of the reservoir which includes unintegrated, integrated and 2LTR circular DNA. Unfortunately no investigations were made to better define this labile reservoir. Further work should be done on the SIV DNA+ samples which should include integrated DNA and cell associated RNA as a minimum to better understand this very important issue. An assessment of the proportion of intact virus would also be highly informative.

We did not have sufficient cells to assess cellular SIV RNA. We assessed integrated SIV DNA in PBMC, LNMC, and GMMC both early and after the 24 week treatment with ART. Integrated DNA after long-term ART suggested the persistent reservoir. Integrated DNA after long-term ART in CD4+ T cells (mostly T_{CM} cells) in lymph nodes correlated with viral rebound following ART discontinuation. This has been added in Figure 4 and Supplementary Figure S2, and detailed on pages 6-7 of the revised manuscript.

3. Unfortunately, a very short duration of ART was given of only 6 months. Do the authors think that a longer duration may have inhibited viral rebound?

We cannot say if a longer duration of ART (>6 months) would inhibit viral rebound, since those studies were not done. However, based on recent data from Drs. Picker and Lifson (which has been presented at meetings), we think that it is possible that >2 years of ART may have prevented rebound in a subset of animals. This is mentioned on page 9 of the revised manuscript, with permission granted from Dr. Picker to mention their unpublished study. As such, our study describes “seeding” the reservoir rather than “establishing” the reservoir throughout the manuscript.

4. The relationship between AUC following viral rebound and time of ART initiation is interesting, especially because the time to rebound did not vary in the 4 groups. In addition, AUC

after viral rebound correlated with SIV DNA in tissue memory CD4+ T-cells at the time of ART discontinuation. Looking at the raw data in figure 4A, it seems that peak viral load was similar for all groups (except day 0 where there was no viral rebound) and set point also pretty similar, except for one monkey in the day 2 group. I am concerned that this relationship is falsely skewed by monkeys that don't rebound (for whom there was a wide range of SIV DNA in tissue as shown in figure S2). Was there still a relationship between AUC post viral rebound and tissue DNA at the time of ART interruption, if the analysis only included monkeys who rebounded? I wonder if this might be a more meaningful analysis

There was a similar relationship between AUC post viral rebound and tissue DNA at the time of ART interruption if restricted to monkeys that rebounded.

5. An extremely important aspect of the study is understanding why monkeys with detectable DNA don't rebound and further mechanistic work should be performed to understand this.

6. It seems that naïve T cells were infected by 24 hours in both LN and GMMC (figure 3B) and in central memory cells in LN (figure S1). This would imply direct infection of these cells, rather than infection of activated memory cells that revert to a naïve or central memory state.

Furthermore, it seems that these infected cells are less stable in GMMC compared to LN (figure 3 and S1) This should be further discussed in relation to establishment of long lived infection in different T-cell subsets and in different tissue sites.

We show new data that viral rebound correlates with integrated DNA in CD4+ T cells (particularly T_{CM} cells) in lymph nodes at the time of ART discontinuation in Figure 4B of the revised manuscript. We prefer to avoid speculation about stability in GMMC vs LNMC, as we don't have sufficient data to be conclusive, and the GMMC data is particularly prone to sampling limitations for only colorectal biopsies.

Discussion

1. On page 9, the authors talk about "initial foci of virus in distal tissues". I was unclear what this meant. If there were a subset of monkeys in whom DNA was detected in tissue but there was no viral rebound, this study provided the opportunity to better characterise this labile reservoir.

We have deleted this sentence on page 10 of the revised manuscript.

2. The concluding statement in relation to PEP is misleading and should be removed from the manuscript.

As suggested, we have removed the comparisons with PEP in humans.

Reviewers' Comments:

Reviewer #2:

Remarks to the Author:

The authors have fully addressed all my previous concerns. No further concerns or problems are detected after the revision of the revised manuscript.

Reviewer #3:

Remarks to the Author:

All concerns have been adequately addressed